# A Review of Sustainable Supplier Selection with Decision-Making Methods from 2018 to 2022

Ömer Karakoç [1,2,*] , Samet Memiş [3] and Bahar Sennaroglu [1]

1   Department of Industrial Engineering, Faculty of Engineering, Marmara University, İstanbul 34854, Türkiye; bsennar@marmara.edu.tr
2   Department of Industrial Engineering, Faculty of Engineering, İstinye University, İstanbul 34396, Türkiye
3   Department of Marine Engineering, Faculty of Maritime, Bandırma Onyedi Eylül University, Balıkesir 10200, Türkiye; samettmemis@gmail.com
*   Correspondence: omerkarakoc123@gmail.com

**Abstract:** Sustainable supplier selection (SSS) is an essential part of the decision-making process in sustainable supply chains. Numerous research studies have been conducted using various decision-making methods to attend to this research-worthy issue. This literature review presents a comprehensive SSS analysis focusing on social, economic, and environmental aspects. The present study spans five years (2018–2022) and considers 101 papers. It provides a detailed breakdown of the papers based on their dates of publication, the countries of the writers, application fields, and journals, and it categorizes them based on their approaches. In addition, this review examines the use of single- or hybrid-form methodologies in the papers reviewed. It also identifies that the TOPSIS, AHP, VIKOR, BWM, DEA, DEMATEL, and MULTIMOORA methods and their extensions are the most frequently used methods in SSS studies. It is concluded that hybrid approaches and their rough, grey, and fuzzy extensions are used to solve real-world problems. However, state-of-the-art mathematical tools, such as soft sets and their hybrid versions with fuzzy sets, have not been utilized in SSS studies. Therefore, this study inspires and encourages the use of such tools in SSS research.

**Keywords:** sustainable supplier selection; supply chain management; multi-criteria decision making; multi-objective decision making; multi-attribute decision making





## 1. Introduction

The motivation for the emergence of sustainability is due to crucial phenomena such as environmental pollution, natural disasters, the depletion of natural resources, greenhouse effects, and global warming. The World Economic Development Commission (WCED) discussed sustainable development in Our Common Future (Brundtland) report [1]. Accordingly, the WCED stated that sustainable development requires the simultaneous adaption of environmental, economic, and social principles. These principles have shaped the policies of companies that want to trade globally. Therefore, companies have featured sustainability principles in their policies and business activities, such as supplier selection in the supply chain, to be competitive. The choice of an effective supplier affects the competitiveness and prestige of the company. Companies recognize that their choice of suppliers can significantly impact their overall sustainability performance. By selecting suppliers who align with their sustainability principles, companies can ensure that their supply chain operations are environmentally responsible, economically viable, and socially equitable. This enhances their competitiveness in the global market and contributes to their reputation as a socially responsible organizations. Consequently, factoring in sustainability principles in supplier selection has become integral to companies' efforts to maintain a competitive edge and uphold their commitment to sustainable development.

Many qualitative and quantitative factors should be considered in sustainable supplier selection (SSS). These multiple factors are generally conflicting, and many alternatives exist

in selecting the appropriate supplier. Therefore, multi-criteria decision-making (MCDM) techniques involve methods and approaches to achieve the best solution in view of the multiple conflicting criteria in SSS. MCDM techniques provide a systematic and structured approach for evaluating suppliers based on various criteria, such as environmental impact, social responsibility, and economic viability. These techniques help companies make informed decisions that align with their sustainability goals and ensure long-term success in supplier selection. This study reviewed the literature for recent SSS studies from 2018 to 2022 to enlighten the aforesaid issue. Within this context, we have conducted a thorough analysis of various papers and categorized them based on several parameters, such as the year of publication, the research methods employed, and the application fields. This comprehensive approach has allowed us to understand the subject matter better and to identify patterns and trends within the research landscape. The major contributions of the review study are summarized as follows:

- The MCDM methods employed for SSS in the last five years have been classified in detail.
- The mathematical concepts frequently utilized in MCDM methods have been identified and discussed in detail.
- It is explained why specific academic journals tend to publish an excessive number of studies on this topic.

The rest of the present study is structured as follows: Section 2 presents the study's research methodology and provides the related review studies on SSS in recent years. Section 3 consists of a descriptive and graphical analysis of the review. Section 4 discusses the results. Finally, the present paper concludes with further research directions.

## 2. Material and Methods

### 2.1. Research Methodology

The present study reviews the literature for recent SSS studies from 2018 to 2022. A literature review requires determining, investigating, and analyzing published papers to answer specific research questions. The structures and gaps of the current literature should be analyzed to identify potential research areas for future work. This literature review considers MCDM-based approaches to SSS. We created a methodology based on the answers to the below research questions:

**Q1:** How can the studies on SSS with social, economic, and environmental dimensions in the five years (2018–2022) be categorized according to MCDM methods?

**Q2:** What extensions are utilized to improve MCDM methods in SSS?

The review study analyzes 101 articles in the Web of Science (WoS) and Scopus databases in detail. The articles are categorized according to their methods and problem types. The techniques employed in the articles are discussed in terms of whether they are of a single or hybrid form. In addition, the articles are classified and listed concerning the years, authors' countries, application areas, and journals. Figure 1 shows a brief overview of the research procedures employed in the present study.

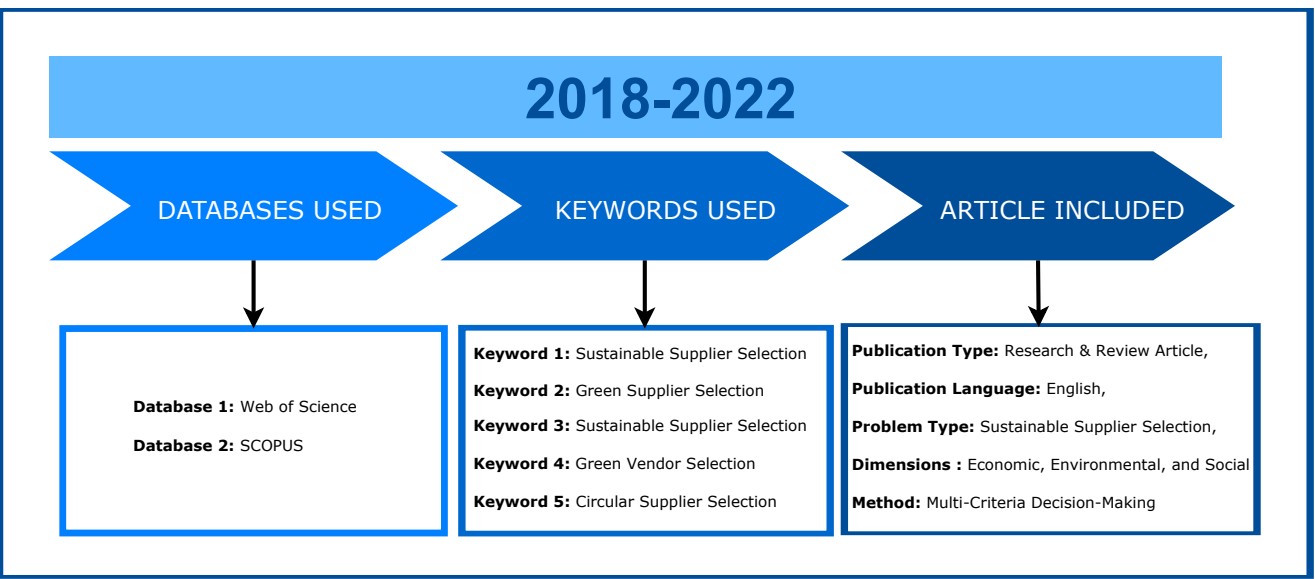

**Figure 1.** Procedure of the present review study.

*2.2. Related Studies*

Various literature review studies have been done on SSS in recent years. Ref. [2] reviewed articles from 1990 to 2018 on SSS's economic, social, and environmental factors. They defined SSS as the selection of appropriate suppliers by considering three dimensions of sustainability to enhance strategic partnerships and to assist in the purchasing process. Ref. [3] conducted an in-depth analysis of 286 articles published between 2002 and 2016 using the Scopus database. The reviewed literature were categorized based on various factors, including the year of publication, the journal they appeared in, the research methodology and design, the mathematical tools and techniques utilized, the data analysis techniques employed, the industry sector studied, the MCDM methods used, and the authors of the articles. Ref. [4] presented a literature review that used the Scopus and Google Scholar databases for supplier selection approaches developed over the past three decades (1990–2019), and 82 relevant articles were identified. They sorted the articles by journal, year, method, and application area. Ref. [5] performed an extensive literature review for 1994–2022. They employed Publish or Perish and VOSviewer to look for authors, citations, journals, word hierarchy maps, and keywords. They operationalized the keywords "green procurement" and "supplier selection" and found 220 articles. Ref. [6] categorized articles based on their problem types and methods. The methods used in the articles were analyzed to determine if they were of single or hybrid form. Furthermore, the papers were organized and listed by year, author's country, application areas, and journals. Table 1 details the review studies for SSS.

When focusing on three dimensions of sustainability, i.e., economic, environmental, and social, only 101 papers met these conditions within the specific period of 2018–2022. We prefer these databases because the WoS and Scopus, the two main bibliographic databases, are the most widely used in academic research [7].

The following issues are addressed in the descriptive analysis part regarding the Supplier Selection Problem:

- The MCDM methods used and their classification;
- The countries of the authors;
- The most frequently used journals;
- The application area.

**Table 1.** Literature review studies on SSS

| Paper | Years Covered | Keywords Searched | Number of Papers Analyzed | Dimensions | Databases Searched |
|---|---|---|---|---|---|
| [3] | 2002–2016 | "sustainable supply chain" | 286 | All dimensions | "Scopus" |
| [2] | 1990–2018 | All combinations of the following groups: Group 1: "purchaser ", "supplier", "vendor", "contractor", "buyer", "seller" Group 2: "purchasing", "evaluation", "assessment", "selection", "procuring", "buying" and "procurement" Group 3: "sustainability", "ecological", "environmental", "green", "social", "corporate social responsibility", "sustainable "and "CSR (Corporate social responsibility)". | 66 | All dimensions | "WoS and Scopus" |
| [4] | 1990–2019 | "sustainable vendor selection" OR "sustainable supplier selection" AND ("*criteria*" OR "*attribute*" OR "multi-objective*" OR "multi*objective*" | 82 | All dimensions | "Google Scholar" and "Scopus" |
| [5] | 1994–2022 | "green procurement" and "supplier selection" | 220 | All dimensions | "WoS" and "Google Scholar" |
| [6] | 2010–2022 | "data envelopment analysis", "supplier", and "sustainable" | 87 | All dimensions | "Scopus" and "WoS" |
| Present study | 2018–2022 | "sustainable supplier selection", "green supplier selection", "sustainable vendor selection", and "green vendor selection" | 101 | All dimensions | "Scopus" and "WoS" |

## 3. Descriptive Analysis

### 3.1. Distribution of Papers Concerning Applied Methods

MCDM problems typically involve several criteria that often in conflict with each other. There are two types of MCDM problems: multi-objective decision making (MODM) and multi-attribute decision making (MADM). Classification is based on the number of alternatives, i.e., if there are infinite alternatives, they are considered MODM problems. On the other hand, if there are a finite number of alternatives, they are classified as MADM problems. When MCDM problems rely on specific rule-based analyses, they are called Function Free models [8].

About 76% of the 101 studies included MADM problems, about 20% included MODM problems, and the remaining were Function Free models.

MCDM problems can be divided into two categories based on the number of decision makers: single decision-making problems (briefly decision-making problems) and group decision-making (GDM) problems. In this review analysis, 67% of the 101 studies were related to single decision making, and the remaining involved GDM problems.

In this review analysis, MCDM methods were handled as single and hybrid methods, and the articles were classified according to the use of single or hybrid methods. Single methods are generally considered classical MCDM methods and their extensions. Figures 2 and 3 present the number of single methods used and their extensions, respectively. Figure 2 shows that the most frequently used single forms were Function Free methods, TOPSIS, and DEA. In addition, Figure 3 shows that the most commonly used extension forms were AHP, TOPSIS, and BMW. Since TOPSIS offers ease of computation and flexibility in application areas, it is widely preferred by decision makers.

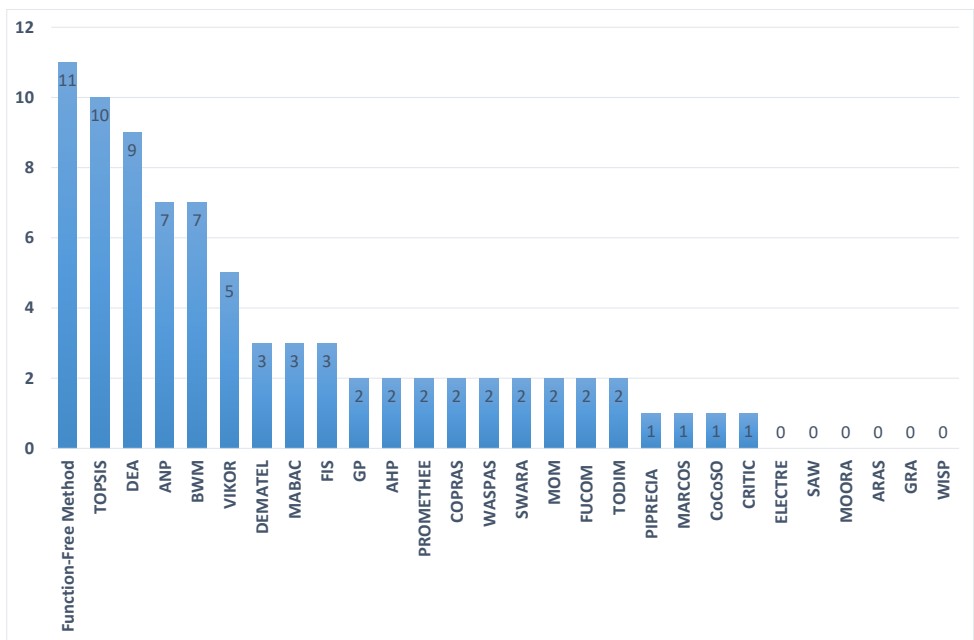

**Figure 2.** Number of single-form MCDM methods.

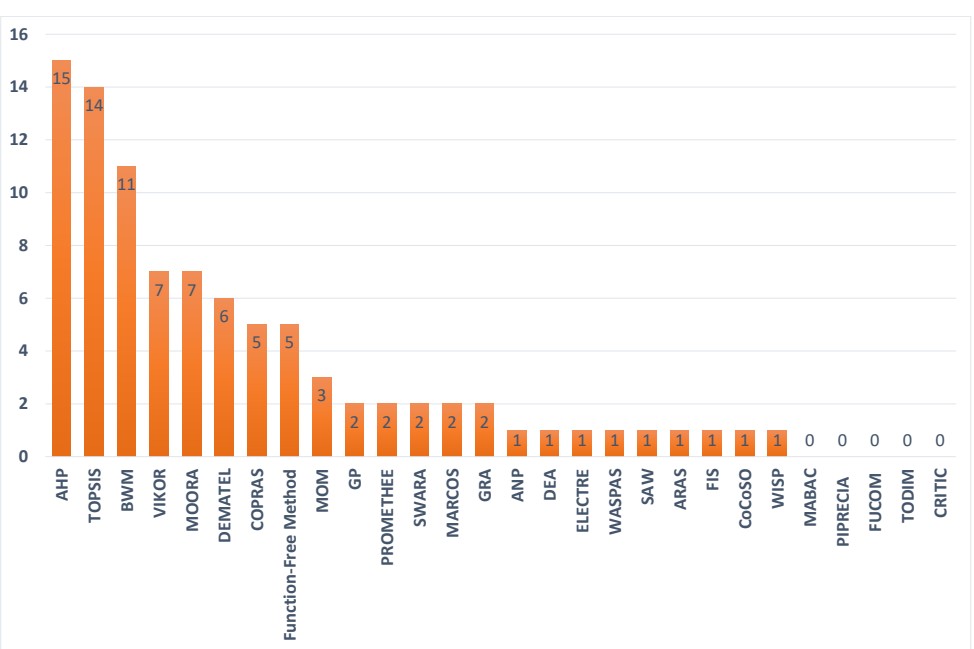

**Figure 3.** Number of extension-form MCDM methods.

A total of 60% of the studies employed hybrid methods, while the remaining used only a single method. The type of problem was determined by the study objectives. It was found that 13 of the studies relied on order allocation and supplier selection. It was identified that TOPSIS, AHP, VIKOR, BWM, DEA, DEMATEL, and MULTIMOORA methods and their extensions were the most used methods to address SSS problems. TOPSIS was used with other techniques as a hybrid method 16 times (Table 2). It was noted that the most used hybrid methods following TOPSIS were BWM (14), AHP (11), VIKOR and DEMATEL (9), COPRAS (7), DEA (6), and MOORA (4) (Tables 2 and 3). These frequently used classical methods have many extensions, such as Fuzzy, Rough, and Grey (Table 4). Figure 4 summarizes the MCDM methods and their extensions in detail, and Tables 2 and 3 briefly explain their extensions and hybrid versions.

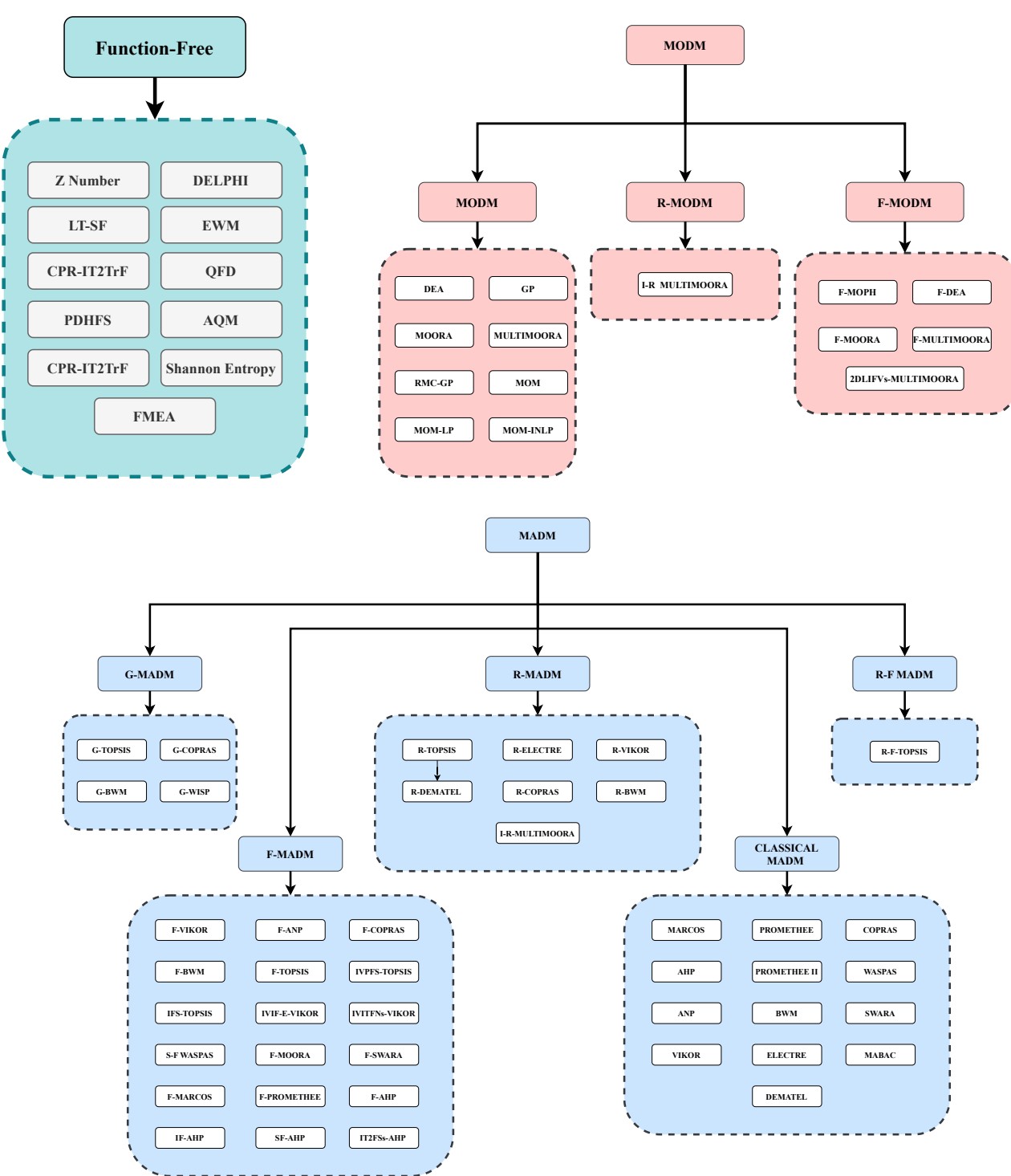

**Figure 4.** Classification of the MCDM methods as free function, MODM, and MADM.

Regarding MCDM methods, one of the most commonly used approaches is TOPSIS [9], which evaluates depending on the distance of the alternative to the ideal solution. Refs. [10–18] used conventional TOPSIS method to select SSS. Refs. [19–24] utilized the Fuzzy variation of TOPSIS (F-TOPSIS) to rank SSS. Ref. [25] introduced the Rough Cloud TOPSIS (R-TOPSIS) method as an extended version of TOPSIS to assess SSS. Ref. [26] proposed Rough-Fuzzy TOPSIS (R-F TOPSIS) approaches for SSS. Ref. [27] recommended Grey-Based TOPSIS (G-TOPSIS) to address SSS problems. Ref. [28] provided an Intuitive-Fuzzy TOPSIS (IFS-

TOPSIS) method to choose SSS. Ref. [29] proposed an extended TOPSIS method based on the concept of interval-valued Pythagorean fuzzy sets (IVPFS-TOPSIS) to evaluate SSS.

The next method employed AHP [30] and its extensions, which was an MCDM method that relies on the subjective pairwise comparisons of multiple criteria in a hierarchical system. Refs. [31,32] used the classical AHP method to evaluate SSS. Refs. [14,20,22,33–35] suggested a Fuzzy extension of AHP (F-AHP) to rank SSS. Ref. [36] provided the Intuition Fuzzy AHP (IF-AHP) for SSS. Refs. [37,38] utilized the Spherical Fuzzy AHP (SF-AHP) method to select SSS. Ref. [39] suggested a variation of the AHP method based on Interval Type-2 Fuzzy sets (IT2FSs-AHP) to rank sustainable suppliers.

VIKOR [40] and its variations are were in the following studies. The VIKOR method provides a ranking list of alternatives by focusing on closeness to the ideal solution and gives a compromise solution with an advantage rate. Refs. [12,41–43] used traditional VIKOR approaches for SSS. Refs. [23,44] provided Fuzzy VIKOR (F-VIKOR) to evaluate SSS. Ref. [45] proposed integrating decision-theoretic rough sets with VIKOR (R-VIKOR) to select appropriate SSS. Ref. [46] utilized the Interval-VIKOR (I-VIKOR) techniques to choose and rank SSS. Ref. [47] suggested the Fuzzy Entropy VIKOR (F-E-VIKOR) method for SSS. Ref. [48] offered a new model, namely Interval-Valued Intuitionistic Fuzzy sets that extended VIKOR (IVIF-E-VIKOR), to evaluate the alternatives (sustainable suppliers). Ref. [49] proposed VIKOR, which was integrated with interval-valued intuitionistic trapezoidal fuzzy numbers (IVITFNs-VIKOR) to select and rank SSS.

The Best-Worst Method (BWM) [50] provides a pairwise comparison system based on comparing the best criterion to other criteria (best-to-others) and all the other criteria to the worst criterion (other-to-worst). Some BMWs and extended approaches were used for SSS. Refs. [11,15,18,51–54] adopted conventional BMW approaches to determine the appropriate SSS. Refs. [21,46,55,56] proposed SSS problems in Fuzzy BMW (F-BMW). Ref. [57] introduced the Rough BWM (R-BWM) method to select SSS. Ref. [58] proposed a Grey BWM (G-BWM) to select SSS. Ref. [59] improved the Best-Worst Method by integrating the concept of interval-valued intuitionistic uncertain linguistic sets (IVIULS-BWM).

Data Envelopment Analysis (DEA) [60], a linear programming technique, assesses performance based on several inputs and outputs from homogeneous decision-making units. Ref. [61] used imprecise data based on goal programming (GP) to choose suppliers in the SSS context. Ref. [62] suggested a solution for the robust SSS and efficient supply network problem by utilizing a Multi-Objective Mixed-Integer Non-Linear Programming (MOMINLP) and DEA approach. Ref. [63] proposed a Fuzzy DEA (F-DEA) model for determining sustainable supplier performance in the car industry. Ref. [64] combined the Fuzzy Decision-Making Trial and Evaluation Laboratory (F-DEMATEL), Analytic Network Process (ANP), and DEA methods to choose a sustainable supplier in the petroleum industry. Ref. [38] addressed an SSS problem in the steel industry with approaches such as Data Envelopment Analysis (DEA), SF-AHP, and global fuzzy-weighted aggregate product evaluation (SF-WASPAS). Ref. [65] proposed hybrid methods such as DEA and Full Consistency Method (FOCUM)–Multi-Attribute Border Approximation Area Comparison (MABAC) to evaluate suppliers in auto-making companies.

The Decision-Making Trial and Evaluation Laboratory (DEMATEL) [66] approach is a widely used method in the field of MCDM. It is employed to construct and analyze a comprehensive structural model that showcases the intricate causal relationships between various factors. This model helps decision makers understand the complex interdependencies among different elements and enables them to make informed and effective decisions based on this piece of knowledge. One of its primary purposes is to verify the dependencies between variables. Another primary purpose is to indicate the overall influence of a given factor on other factors. Some DEMATEL and extended approaches are used for SSS. Refs. [41,43,67] used conventional DEMATEL approaches to determine appropriate SSS. Ref. [44] suggested combining the Rough DEMATEL (R-DEMATEL) and F-VIKOR methods to select a SSS.

**Table 2.** VIKOR, TOPSIS, AHP, and BWM MCDM methods and their extensions, as well as hybrid versions, from 2018 to 2022.

| Methods | Extensions | Hybrids | References |
|---|---|---|---|
| VIKOR | VIKOR | DEMATEL + VIKOR | [41] |
| | | VIKOR + ANP | [12] |
| | | IVIF-DEMATEL + ANP + VIKOR | [42] |
| | | DEMATEL + ANP + VIKOR | [43] |
| | F-VIKOR | F-VIKOR + R DEMATEL | [44] |
| | | F-VIKOR + F-TOPSIS | [23] |
| | R-VIKOR | | [45] |
| | I-VIKOR | F-BWM + I-VIKOR | [46] |
| | F-E-VIKOR | | [47] |
| | IVIF-E-VIKOR | IVIF-E-VIKOR + IVIF-MARCOS | [48] |
| | IVITFNs-VIKOR | IVITFNs-QFD + IVITFNs-VIKOR | [59] |
| TOPSIS | TOPSIS | TOPSIS + MOM | [10] |
| | | BWM + TOPSIS + WASPAS | [11] |
| | | TOPSIS + ANFIS | [16] |
| | | ANP + TOPSIS | [12] |
| | | F-AHP + TOPSIS + F-MOPM | [20] |
| | | TOPSIS + ELECTRE | [13] |
| | | F-AHP + TOPSIS | [17] |
| | | F-GRA + BWM + TOPSIS | [18] |
| | | BWM + TOPSIS | [15] |
| | F-TOPSIS | Delphi + F-TOPSIS + GP | [19] |
| | | F-AHP + F-TOPSIS + MOOM | [20] |
| | | F-COPRAS + F-MULTIMOORA + F-TOPSIS + F-BWM | [21] |
| | | F-AHP + F-TOPSIS + F-MOPH | [22] |
| | | F-VIKOR + F-TOPSIS | [23] |
| | | ISM + F-DEMATEL + ANP + F-TOPSIS | [24] |
| | R-TOPSIS | | [25] |
| | R-F TOPSIS | R-F-DEMATEL + R-F-TOPSIS | [26] |
| | G-TOPSIS | | [27] |
| | IFS-TOPSIS | | [28] |
| | IVPFS-TOPSIS | | [29] |
| AHP | AHP | AHP + FIS | [31] |
| | | AHP + F-MULTIMOORA | [32] |
| | IT2FSs-AHP | | [39] |
| | F-AHP | F-AHP + F-COPRAS | [33] |
| | | F-AHP + F-TOPSIS + MOOM | [20] |
| | | F-AHP + TOPSIS + F-MOPM | [22] |
| | | F-AHP + PROMETHEE | [34] |
| | | F-AHP + F-TOPSIS | [14] |
| | | F-AHP + TOPSIS | [35] |
| | IF-AHP | IF-AHP + TODIM | [36] |
| | SF-AHP | SF-AHP + G-COPRAS | [37] |
| | | DEA + SF-AHP + SF-WASPAS | [38] |
| BWM | BWM | BWM + RMC-GP | [51] |
| | | BWM + F-Shannon Entropy + F-MULTIMOORA | [52] |
| | | BWM + MULTIMOORA | [54] |
| | | BWM + 2DLIFVs-MULTIMOORA | [53] |
| | | BWM + TOPSIS + WASPAS | [11] |
| | | BWM + F-GRA + TOPSIS | [18] |
| | | BWM + TOPSIS | [15] |
| | F-BWM | F-BWM + CoCoSo'B | [36] |
| | | F-BWM + I-VIKOR | [46] |
| | | F-BWM + F-DEMATEL + F-ANP + FIS | [56] |
| | | F-BWM + F-COPRAS + F-MULTIMOORA + F-TOPSIS | [21] |
| | R-BWM | R-BWM + MABAC | [57] |
| | G-BWM | G-BWM + G-WISP | [58] |
| | IVIULSs-BWM | IVIULSs-BWM + IVIULSs-AQM | [49] |

**Table 3.** DEA, DEMATEL, MOORA and COPRAS MCDM methods and their extensions, as well as hybrid versions, from 2018 to 2022.

| Methods | Extensions | Hybrids | References |
|---|---|---|---|
| DEA | DEA | GP + DEA | [61] |
| | | MOM-INLP + DEA | [62] |
| | | F-AHP + DEA | [68] |
| | | F-DEMATEL + ANP + DEA | [64] |
| | | DEA + FUCOM + MABAC | [65] |
| | | DEA + SF-AHP + SF-WASPAS | [38] |
| | F-DEA | | [63] |
| DEMATEL | DEMATEL | DEMATEL + ANP + VIKOR | [43] |
| | | F-GRA + FMEA + EWM + DEMATEL | [67] |
| | | DEMATEL + VIKOR | [41] |
| | R-DEMATEL | R-DEMATEL + F-VIKOR | [44] |
| | R-F DEMATEL | R-F-DEMATEL + R-F-TOPSIS | [26] |
| | F-DEMATEL | ISM + F-DEMATEL + ANP + F-TOPSIS | [24] |
| | | F-DEMATEL + ANP + DEA | [64] |
| | | F-DEMATEL + F-BWM + F-ANP + FIS | [56] |
| | IVIF-DEMATEL | IVIF-DEMATEL + ANP + VIKOR | [42] |
| MOORA | F-MOORA | | [69] |
| | MULTIMOORA | BWM + MULTIMOORA | [54,70] |
| | I-R-MULTIMOORA | | [53] |
| | F-MULTIMOORA | F-COPRAS + F-MULTIMOORA + F-TOPSIS + F-BWM | [21] |
| | | AHP + F-MULTIMOORA | [32] |
| | | BWM + F-Shannon Entropy + F-MULTIMOORA | [52] |
| | 2DLIFVs-MULTIMOORA | BWM + 2DLIFVs-MULTIMOORA | [54] |
| COPRAS | COPRAS | SWARA + COPRAS | [71] |
| | | HF-SWARA + COPRAS | [72] |
| | F-COPRAS | F-COPRAS + F-MULTIMOORA + F-TOPSIS + F-BWM | [21] |
| | | F-COPRAS + F-AHP | [33] |
| | G-COPRAS | F-Delphi + ISM + G-COPRAS + ANP | [73] |
| | | SF-AHP + G-COPRAS | [37] |
| | R-COPRAS | FUCOM + R-COPRAS | [74] |

**Table 4.** Frequently used MCDM methods and their extensions from the years 2018 to 2022.

| Methods | Extensions | | | | |
|---|---|---|---|---|---|
| | Rough Sets-Based | Fuzzy Sets-Based | | | Grey Set-Based |
| TOPSIS | R-TOPSIS | IVPFS-TOPSIS | F-TOPSIS | IFS-TOPSIS | G-TOPSIS |
| AHP | – | IT2FSs-AHP | SF-AHP | IF-AHP | – |
| BWM | R-BWM | | IVIULSs-BWM | | G-BWM |
| MOORA | I-R MULTIMOORA | 2DLIFVs-MULTIMOORA | F- MOORA | F-MULTIMOORA | |
| DEMATEL | – | IVIF-DEMATEL | | F-DEMATEL | |
| COPRAS | R- COPRAS | | F- COPRAS | | G-COPRAS |
| VIKOR | R-VIKOR | I-VIKOR | F-E-VIKOR | IVITFNs-VIKOR | IVIF-E-VIKOR | – |
| DEA | | | F- DEA | | |

Ref. [26] combined Rough-Fuzzy DEMATEL (R-F-DEMATEL) and R-F-TOPSIS approaches to the SSS problem. Refs. [24,42,56,64] used a fuzzy variation of the DEMATEL (F-DEMATEL) and Interval-Valued Intuitionistic Fuzzy DEMATEL (IVIF-DEMATEL) approaches to assess SSS.

Multi-Objective Optimization by Ratio Analysis (MOORA) is an optimization technique based on ratio analysis. In general, fuzzy variations are used in the SSS. Ref. [69] suggested a ratio-based fuzzy multi-objective optimization model (F-MOORA) be employed to rank SSS. Refs. [32,52,70] proposed a Multi-Objective Optimization by Ratio Analysis with a Full Multiplicative Form (MULTIMOORA) approach to solve a new SSS problem. Ref. [21] used a fuzzy version of MULTIMOORA (F- MULTIMOORA) to select SSS. Ref. [53] extended the MULTIMOORA method using intuitionistic linguistic rough numbers (I-R-MULTIMOORA) for SSS problems. Ref. [54] suggested hybrid multi-attribute GDM methods that depend upon the BWM and MULTIMOORA methods with 2D linguistic intuitionistic fuzzy variables (MULTIMOORA-2DLIFVs) for SSS.

Complex Proportional Assessment (COPRAS) uses a progressive ranking and evaluation procedure for alternatives by calculating relative importance and utility degrees. In the SSS context, refs. [71,72] used the conventional COPRAS method. Refs. [21,33] utilized the fuzzy version of COPRAS (F-COPRAS) to evaluate SSS. Refs. [37,73] introduce a new solution to the SSS problem using Grey COPRAS (G-COPRAS). Ref. [74] recommended a new rough (R-COPRAS) method to evaluate SSS.

### 3.2. Distribution of the Papers Concerning Journal, Area, and Year

The 101 publications from peer-reviewed journals in the WoS and Scopus databases were examined in the context of the present review. The distribution of the analyzed articles by year is shown in Figure 5.

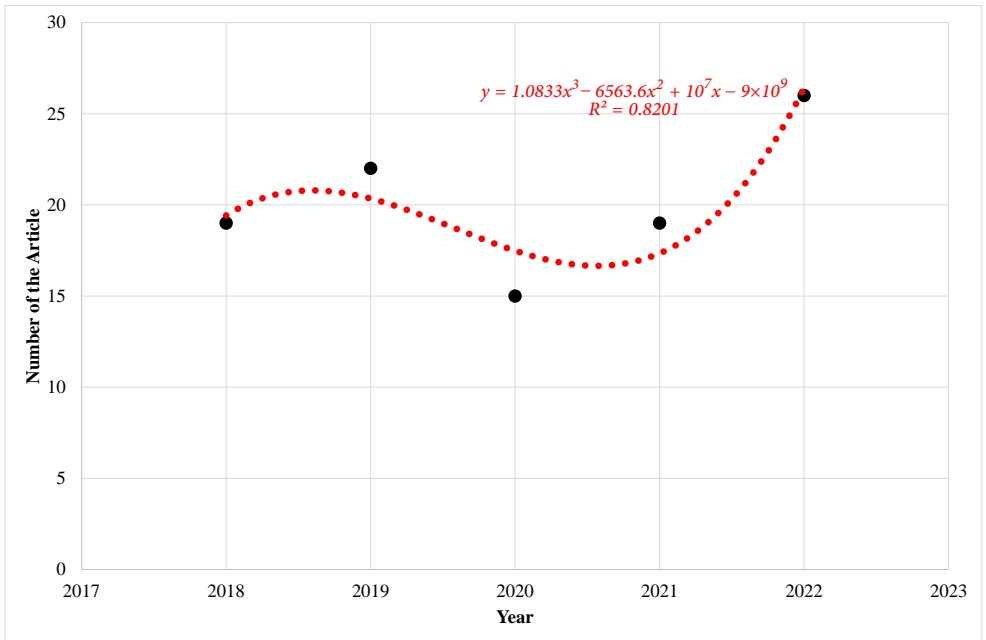

The equation shown in the figure:

$$y = 1.0833x^3 - 6563.6x^2 + 10^7 x - 9 \times 10^9$$
$$R^2 = 0.8201$$

**Figure 5.** The distribution of analyzed articles by year (Black dots) and the polynomial regression model of the distribution (Red dots)

A significant increase was observed on approaches to SSS in scientific articles since 2020. The primary reason for the rise in scientific articles in the last three years can be explained with the various models developed under uncertainty or via the selection of suppliers in fuzzy and rough environments rather than the classical MCDM method approaches.

Based on the findings in Figure 6, the automotive industry was the most frequently researched field in the past five years. This is mainly because the vehicle supplier selection issue is often addressed in research studies, as adopting sustainable vehicles can help reduce carbon emissions and minimize environmental damage. The energy sector is the second most commonly studied field, as the precise selection and use of energy sources are crucial.

The review of 101 articles revealed that China and Iran were the top contributing countries (Figure 7). The reason for this might be the interest of authors from these countries in the field of SSS due to the importance of sustainability for sectors such as transport, energy, and automotive.

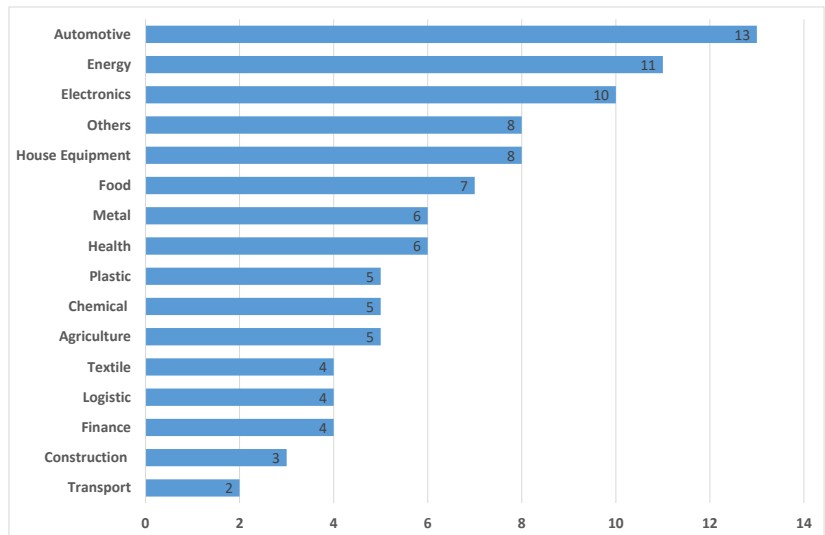

**Figure 6.** Distribution of the analyzed articles according to the application area.

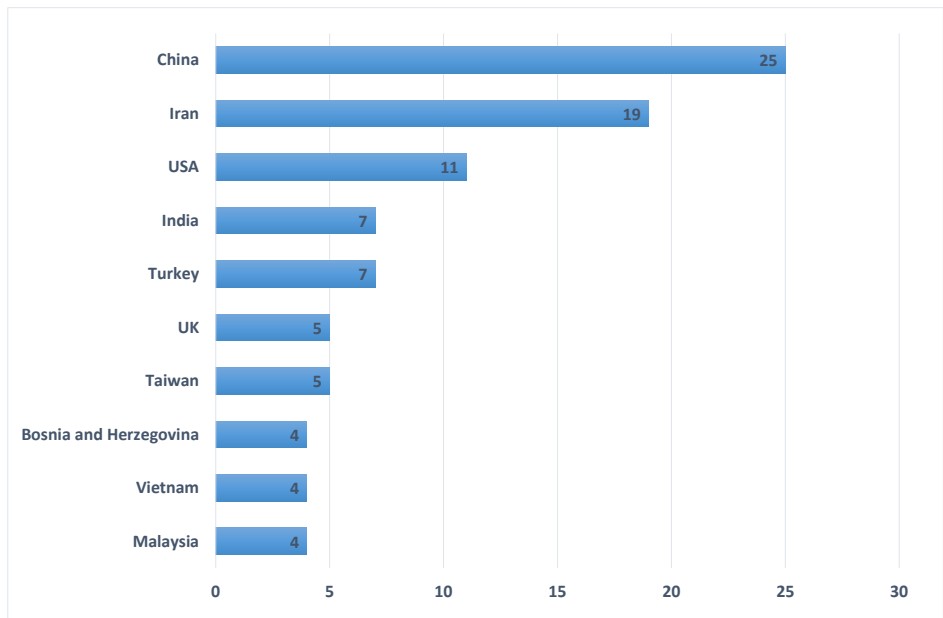

**Figure 7.** Distribution of the analyzed articles according to the author's country.

In the SSS context, the Journal of Cleaner Production and Sustainability were the journals that attracted the most attention from the authors (Figure 8). The reason for this may be that the index is SCI/SCI-E, which is included in the Q1 and Q2 ranks, and that the impact factor is high. Moreover, other reasons could be that "Green & Sustainable Science & Technology" is one of the WoS subjects of the journals and Sustainability is an open-access journal.

According to Figure 8, 49 articles out of the 101 papers were published in 10 different journals related to the SSS context. Table 5 provides information about the access type, the related index, rank status, impact factor score, cited scores of the associated journals, and subject headings. The top 10 journals belong in Q1 and Q2, and they are either open-access or open-access supported. The topics covered in these journals have a wide range, so they have attracted the authors' attention.

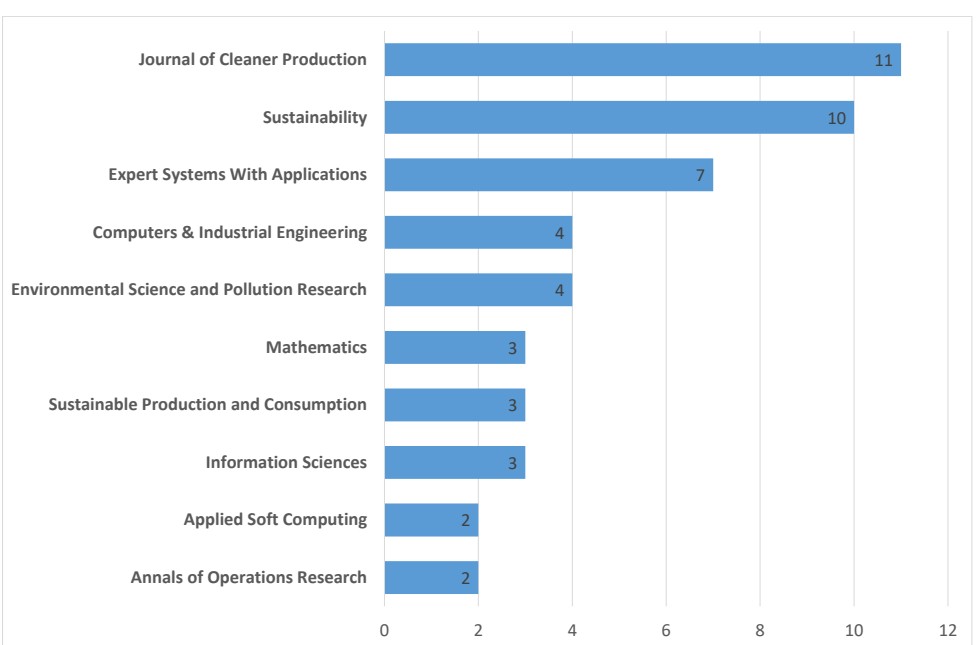

**Figure 8.** Distribution of the analyzed articles according to the journals.

**Table 5.** Details of the ten journals (retrieved from WoS 2022).

| Journal | Access | Index | Rank | Impact Factor | CiteScore | Subject |
|---|---|---|---|---|---|---|
| Journal of Cleaner Production | Subscription Open Access | SCIE | Q1 | 11.072 | 15.8 | "Engineering", "Environmental Sciences" "Green & Sustainable Science & Technology" |
| Sustainability | Open Access | SCIE SSCI | Q2 | 3889 | 5 | "Environmental Sciences" "Environmental Studies" "Green & Sustainable Science & Technology" |
| Expert Systems with Application | Subscription Open Access | SCIE | Q1 | 8.665 | 12.2 | "Current Contents Engineering, Computing & Technology" "Essential Science Indicators" |
| Computers Industrial Engineering | Subscription Open Access | SCIE | Q1 | 7.18 | 9.7 | "Computer Science", "Interdisciplinary Applications" "Engineering", "Industrial" |
| Environmental Science and Pollution Research | Subscription Open Access | SCIE | Q2 | 5.190 | 5.5 | "Environmental Sciences" |
| Mathematics | Open Access | SCIE | Q1 | 2.592 | 2.9 | "Pure and Applied Mathematics" |
| Sustainable Production and Consumption | Open Access | SCIE SSCI | Q1 | 8.921 | 8.1 | "Current Contents Engineering, Computing & Technology" "Current Contents Social and Behavioral Sciences" "Essential Science Indicators" |
| Information Sciences | Subscription Open Access | SCIE | Q1 | 8.233 | 12.1 | "Computer Science", "Information Systems" |
| Applied Soft Computing | Subscription Open Access | SCIE | Q1 | 8.263 | 12.4 | "Computer Science", "Interdisciplinary Applications" "Computer Science", "Artificial Intelligence" |
| Annals of Operation Research | Subscription Open Access | SCIE | Q2 | 4.820 | 5.2 | "Operations Research & Management Science" |

## 4. Discussion

This literature review study retrieved articles from the WoS and Scopus databases about SSS between 2018 and 2022. The articles were categorized based on their problem

types and methods. The methods were classified as a single or hybrid form. Furthermore, the articles were organized and listed by year, author's country, application areas, and journals.

In their research, the authors of [3,4] concluded that DEA, TOPSIS, AHP, and their variations are the most frequently utilized methods. Our study supports these findings and additionally covers new extensions of these methods in terms of rough, grey, and fuzzy versions. The present study finds that China is the first country with the most publications on SSS, just as the study of [5] also concluded.

This study shows that the most utilized methods were TOPSIS, AHP, VIKOR, BWM, DEA, DEMATEL, and MULTIMOORA, including their extensions regarding rough, grey, and fuzzy versions. In addition, since reducing carbon footprints in supply chains has become crucially important, it has been observed that SSS studies use MCDM methods to minimize environmental impact, especially in the automotive, energy, and electronics fields (Figure 8). One of the significant reasons therein is that the authors tend to utilize classical methods and their fuzzy extensions because of their ease of implementation. Another is that their demonstrated effectiveness in tackling complex decision problems across various industries, such as automotive, electronics, and energy, may play a pivotal role in SSS problems. On the other hand, decision makers' evaluations when using MCDM methods are generally subjective, thus requiring the implementation of fuzzy methods and their extensions, which are complex, difficult to compute, and have many different types for selection. Hybrid methods are more successful than single methods but are more complex. To deal with real-world problems, various fuzzy-based structures such as Interval-Valued Intuitionistic Fuzzy, Spherical Fuzzy, Intuitive-Fuzzy, Rough-Fuzzy, Fuzzy Entropy, Interval-Valued Intuitionistic trapezoidal Fuzzy, and global fuzzy weighted aggregate product evaluation have been suggested. Moreover, this study highlighted that soft sets and their extensions, though encompassing all three dimensions of sustainability, have not been utilized in SSS studies.

In contrast to fuzzy sets, soft sets—which are obtained from parameterizing the considered set—provide convenience in constructing functions without employing complicated membership functions [75]. To deal with problems containing fuzzy parameters or alternatives (objects), the hybrid versions of fuzzy sets and soft sets, such as fuzzy soft sets [76,77], fuzzy parameterized soft sets [78], and fuzzy parameterized fuzzy soft sets (*fpfs*-set) [79], have been introduced. The concept of *fpfs*-sets has been prominent among others due to its modeling ability. However, if a problem with several criteria and high uncertainty arises, computerizing the *fpfs*-sets is compulsory. To this end, fuzzy soft matrices and fuzzy parameterized fuzzy soft matrices (*fpfs*-matrices) have been offered.

Despite their modeling ability, soft set-based structures, such as *fpfs*-matrices, have no real applications to SSS. There are only three soft set-based empirical studies on SSS [80–82]. In light of the aforementioned results, future studies of SSS should focus on the application of SDM via *fpfs*-matrices. In addition, accessible MATLAB codes, in such repositories as GitHub, MathWorks, etc., of the SDM methods constructed and configured by *fpfs*-matrices offer remarkable advantages for researchers who intend to utilize them for SSS.

## 5. Conclusions

In this paper, we utilized a comprehensive search strategy to conduct a research review on SSS. Specifically, we searched for articles using the keywords "sustainable vendor selection", "green supplier selection", "sustainable supplier selection", and "green vendor selection" within the WoS and Scopus databases. In total, our search yielded 101 articles.

We systematically classified the obtained 101 papers based on the methods and the number of the decision makers involved. The tables and figures presented the descriptive analyses. Approximately 76% of the 101 studies relied on MADM, about 20% of the studies used MODM, while the remaining used Function Free Models. A total of 67% of the 101 studies were related to single decision making, and the remaining were related to

GDM problems. The articles were classified according to single methods (40%) and hybrid methods (60%). It was observed that hybrid methods were preferred to single methods. TOPSIS and BWM were mostly chosen for the hybrid techniques, followed by AHP, VIKOR, DEMATEL, COPRAS, DEA, and MOORA. On the other hand, hybrid methods with their rough, grey, and fuzzy extensions (Table 4) were utilized to solve real problems. Table 4 shows that fuzzy sets-based MCDM methods have been commonly used.

Sustainability and the Journal of Cleaner Production tend to publish a great many studies on SSS due to their high impact factor and because "Green & Sustainable Science & Technology" is one of their WoS subjects. In addition, due to being an open-access journal, Sustainability saw increases in the number of article published by it.

The present review study revealed that soft sets [75] and their extensions, i.e., fuzzy parameterized fuzzy soft sets/matrices [79,83], intuitionistic fuzzy parameterized intuitionistic fuzzy soft sets/matrices [84–86], interval-valued intuitionistic fuzzy parameterized interval-valued intuitionistic fuzzy soft sets/matrices [87,88], and picture fuzzy parameterized picture fuzzy soft sets/matrices [89,90], have been not yet used in SSS problems that include all three dimensions of sustainability. Therefore, soft sets-based concepts, being prominent through their modeling skill, can be applied to SSS problems for further research.

As a consequence of this review study, by examining articles retrieved from the WoS and Scopus databases between 2018 and 2022, researchers can better understand the current state of research and the emerging trends in this field. They can contribute to advancing knowledge in SSS studies and be informed about future research directions and methodologies. Therefore, this study is considered a significant contribution to the field of sustainable supply chains.

**Author Contributions:** B.S. and S.M. supervised this study's findings. B.S. and Ö.K. devised the main conceptual ideas and developed the theoretical framework. Ö.K. gathered the data. Ö.K. wrote the manuscript with support from B.S. and S.M. Ö.K. and S.M. analyzed the data and presented the visual results. S.M. and B.S. reviewed and edited the paper. This paper was derived from the first author's doctoral dissertation and was supervised by B.S. and S.M. All the authors discussed the results and contributed to the final paper. All authors have read and agreed to the published version of the manuscript.

**Funding:** This research received no external funding.

**Institutional Review Board Statement:** Not applicable.

**Informed Consent Statement:** Not applicable.

**Data Availability Statement:** The data that support the findings of this study are available from the corresponding author upon reasonable request.

**Conflicts of Interest:** The authors declare that they have no known competing financial interests or personal relationships that could have appeared to influence the work reported in this paper.

## Abbreviations

The reviewed research studies for the SSS problems related to the concepts and MCDM methods.

| Concept and Method | Abbreviation | Reference for SSS |
|---|---|---|
| Multi Criteria Decision Making | MCDM | |
| Goal Programming | GP | [91] |
| Fuzzy Goal Programming | F-GP | [92] |
| Revised Multi-Choice Goal Programming | RMC-GP | [51] |
| VIseKriterijumsa Optimizacija I Kompromisno Resenje | VIKOR | [93] |
| Fuzzy VIKOR | F-VIKOR | [23,44] |
| Rough VIKOR | R-VIKOR | [45] |

| | | |
|---|---|---|
| Interval VIKOR | I-VIKOR | [46] |
| Fuzzy Entropy VIKOR | F-E-VIKOR | [47] |
| Interval-Valued Intuitionistic Fuzzy sets Extended VIKOR | IVIF-E-VIKOR | [48] |
| Integrated with Interval-Valued Intuitionistic Trapezoidal Fuzzy Numbers VIKOR | IVITFNs-VIKOR | [49] |
| Technique for Order Preference by Similarity to Ideal Solution | TOPSIS | [10,16,18,94] |
| Fuzzy TOPSIS | F-TOPSIS | [14,19,21,23,24,95] |
| Rough Cloud TOPSIS | R-TOPSIS | [28] |
| Rough-Fuzzy TOPSIS | R-F TOPSIS | [26] |
| Grey-Based TOPSIS | G-TOPSIS | [27] |
| Intuitive Fuzzy TOPSIS | IFS-TOPSIS | [28] |
| Interval-Valued Pythagorean Fuzzy Set TOPSIS | IVPFS-TOPSIS | [29] |
| Analytic Hierarchy Process | AHP | [31,96,97] |
| Interval Type-2 Fuzzy Sets AHP | IT2FSs-AHP | [36] |
| Fuzzy AHP | F-AHP | [22,33–35,68,94,98–101] |
| Intuition Fuzzy AHP | IF-AHP | [36] |
| Spherical Fuzzy AHP | SF-AHP | [37,38] |
| Analytic Network Process | ANP | [12,24,42,43,64,102] |
| Fuzzy-ANP | F-ANP | [56] |
| Data Envelopment Analysis | DEA | [61,68,103,104] |
| Fuzzy DEA | F-DEA | [65] |
| Preference Ranking Organization Method for Enrichment Evaluation | PROMETHEE | [34] |
| Fuzzy Preference Ranking Organization Method for Enrichment Evaluation | F-PROMETHEE | [101] |
| Preference Ranking Organization Method for Enrichment Evaluation | PROMETHEE II | [105] |
| Elimination and Choice Translating Reality English | ELECTRE | [106] |
| Rough ELECTRE | R-ELECTRE | [106] |
| Decision Making Trial and Evaluation Laboratory | DEMATEL | [43,67] |
| Rough DEMATEL | R-DEMATEL | [44] |
| Rough Fuzzy DEMATEL | R-F DEMATEL | [26] |
| Interval-Valued Intuitionistic Fuzzy DEMATEL | IVIF-DEMATEL | [42] |
| Multi Attribute Border Approximation Area Comparison | MABAC | [57,65,107] |
| Best Worst Method | BWM | [11,51] |
| Fuzzy BWM | F-BWM | [55,56,108–111] |
| Rough BWM | R-BWM | [57] |
| Grey BWM | G-BWM | [58] |
| Interval-Valued Intuitionistic Uncertain Linguistic Sets BMW | IVIULSs-BWM | [59] |
| Complex Proportional Assessment | COPRAS | [71,72] |
| Rough COPRAS | R-COPRAS | [74] |
| Grey COPRAS | G-COPRAS | [37,73] |
| Fuzzy COPRAS | F-COPRAS | [21,33] |
| Weighted Aggregated Sum Product Assessment | WASPAS | [11,112] |
| Global Fuzzy WASPAS | SF-WASPAS | [38] |
| Rough Weighted Sum Model | R-SAW | [113] |
| Fuzzy Multi-Objective Optimization By Ratio Analysis | F-MOORA | [69] |
| The Multi-Objective Optimization by Ratio Analysis with Full Multiplicative Form | MULTIMOORA | [54,70] |
| Intuitionistic-Rough MULTIMOORA | I-R-MULTIMOORA | [53] |
| Fuzzy MULTIMOORA | F-MULTIMOORA | [21,32,52] |
| Two-Dimension Linguistic Intuitionistic Fuzzy Variables MULTIMOORA | 2DLIFVs-MULTIMOORA | [54] |
| Stepwise Weighted Assessment Ratio Analysis | SWARA | [71,112] |
| Fuzzy SWARA | F-SWARA | [114] |
| Hesitant-Fuzzy SWARA | HF-SWARA | [72] |
| Delphi | Delphi | [19,73] |
| Full Consistency Method | FUCOM | [74,113] |

| | | |
|---|---|---|
| Pivot Binary Relative Criterion Importance Assessment | PIPRECIA | [107] |
| Measurement of Alternatives and Ranking According to Compromise Solution | MARCOS | [48,115] |
| Fuzzy MARCOS | F-MARCOS | [114] |
| Interval-Valued Intuitionistic Fuzzy Sets MARCOS | IVIF-MARCOS | [48] |
| Fuzzy Additive Ratio Assessment Method | F-ARAS | [111] |
| Fuzzy Inference Systems | FIS | [56,109,116] |
| Adaptive Neuro FIS | ANFIS | [16] |
| Multi-Objective Optimization Model | MOOM | [20] |
| Multi-Objective Model | MOM | [10] |
| Multi-Objective Linear Programming Model | MOM-LP | [117] |
| Multi-Objective Mixed-Integer Non-Linear Programming | MOM-INLP | [62] |
| Fuzzy KANO | F-KANO | [118] |
| Tomada de Decisão Iterativa Multicritério | TODIM | [36,119] |
| Interval-Valued Intuitionistic Trapezoidal Fuzzy Numbers-Quality Function Deployment | IVITFNs-QFD | [49] |
| Fuzzy Grey Relational Analysis | F-GRA | [18] |
| Failure Mode and Effects Analysis | FMEA | [67] |
| Fuzzy FMEA | F-FMEA | [120] |
| Interval Value Fuzzy Set FMEA | IVF-FMEA | [121] |
| Interval-Valued Intuitionistic Uncertain Linguistic Sets Alternative Queuing Method | IVIULSs-AQM | [59] |
| Fuzzy Multi-Objective Programming Model | F-MOPM | [22] |
| Combined Compromise Solution | CoCoSo | [122] |
| Combined Compromise Solution with Bonferroni | CoCoSo'B | [55] |
| Grey Weighted Sum-Product | G-WISP | [58] |
| Entropy Weight Method | EWM | [67] |
| Fuzzy Shannon Entropy Method | F-SEM | [52] |
| Interpretive Structural Modelling | ISM | [24,73] |
| Z-information Possibilistic Method | Z Number | [123] |
| Linguistic t-Spherical Fuzzy Generalized Distance Measure | LT-SF | [17] |
| Interval Type-2 Trapezoidal Fuzzy Set Complex Preference Information | CPR-IT2TrF | [124] |
| Power Dual Hesitant Fuzzy Setting | PDHFS | [125] |
| Criteria Importance Through Inter-Criteria Correlation | CRITIC | [122] |

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
