# Peer review of "A Review of Sustainable Supplier Selection with Decision-Making Methods from 2018 to 2022"

_sustainability, doi:10.3390/su16010125_

Round 1

Reviewer 1 Report

Comments and Suggestions for Authors

If it is about a comprehensive analysis for Sustainable supplier selection, why did you take into consideration only 101 papers in this specific period? Per my knowledge, there are some missing sources. You mentioned that a total of 780 articles were identified, but you excluded retracted articles, duplicated articles, and articles not related to the SSS problem. You must elaborate on the methodology of the extraction of these articles. E.g, if you mention related articles, what are the criteria by which an article is considered related? Or, when you mention Objectives are not directly the supplier selection problem, you should define a basis or criteria to state what is an objective and what is not an objective directly linked to the supplier selection.

The presentation of the results is done correctly. It is interesting to see in your paper brief presentations of these methods (TOPSIS, VIKOR, AHP…etc), and their approaches.

Discussion is given in a very narrow way. However, to enhance the quality of this discussion section, it would be beneficial to provide a more detailed analysis of the identified trends. For instance, the authors could delve into the reasons behind the popularity of certain methods or extensions and discuss any emerging areas where these methods are being applied most frequently. Furthermore, it would be helpful to offer insights into the limitations or challenges associated with the utilization of these methods, which could guide future research directions.

Author Response

Dear Reviewer,

We have attached the response letter. Please consider it.

Best regards,

Reviewer 2 Report

Comments and Suggestions for Authors

The article handles a very specific theme, focusing in a single argument; This is very good as it allows delving deeply on the issue.

Nonetheless, there are some issues that should be addressed prior of publishing:

1.       The abstracts tells very little, please inform more on the results and implications of the study. What is the benefit your study conveys to the audience?

2.       Each and every falsifiable statement must be anchored by a reference. You have large parts (for instance, the first paragraph, but not only) without any ref.

3.       Introduction must show only the motivation and the context of your study. A research question is lacking. Related studies and table 1 must be reallocated to the next chapter.

4.       Table 1 and other exhibits are hard to read, please enlarge the font, and improve the quality. (see Figure 1, it is OK)

5.       Figures 2 to 4 have little information and are useless, please remove, and replace them by cursive writing in the body of the text.

6.       I believe you should split Figure 5 in two to be more clear. A further explanation the content of the figure is due in the text.

7.       Please strictly follow the author´s guideline for Table 3.

8.       Figure 7 is a very important finding of your work. I believe you should adjust an exponential model to it to evidence how its growing happens.

9.       Figure 8 and 9 should follow the same pattern of Figure 10, from most to less.

10.   Please take a look in the way you employ the word follow (I believe next should be better sometimes).

11.   In the first page you must advice the reader on the abbreviations list at the end.

Comments on the Quality of English Language

Must be reviewed

Author Response

(The authors gave the same response as above.)

Reviewer 3 Report

Comments and Suggestions for Authors

The title is clear, indicating a review of sustainable supplier selection methods over a recent five-year period.

The abstract provides a concise overview, summarising the scope, method, and findings effectively. However, it might benefit from highlighting any novel contributions or significant gaps addressed in the literature.

The introduction sets the stage by discussing the importance of sustainability and its impact on supplier selection.

The rationale for the study is established, though it could be strengthened by explicitly stating the unique contribution of this review in the context of existing literature.

The selection criteria for the literature are clear, but the justification for excluding certain articles (e.g., those not in English) should be elaborated to ensure it doesn’t introduce bias.

The methodological approach is systematic, but the paper could benefit from a discussion on the potential limitations of using only Web of Science and Scopus databases.

The categorization of methods into single and hybrid forms is useful, but the discussion lacks critical evaluation of the pros and cons of these approaches.

The paper compiles recent studies and presents a classification of methods, which is valuable. Yet, it is not immediately clear how this review advances the understanding of sustainable supplier selection beyond compiling information.

Mention of “soft sets” and their absence in the literature is intriguing but requires a more detailed explanation of how they might be used in future SSS studies.

The paper is generally well-written and formatted. However, the consistency in citing (e.g., [number]) could be improved for readability and standardisation.

Author Response

(The authors gave the same response as above.)

Reviewer 4 Report

Comments and Suggestions for Authors

In the paper, the authors presented a review of research works on sustainable supplier selection focusing on social, economic, and environmental aspects. The 101 papers overviewed come from 2018-2022. The articles have been categorized relative to the year of publication, nationality of the writers, application fields, journal of publication, and the applied approaches. Furthermore, the use of single or hybrid methodologies were considered during categorization. The review appears to be quite extensive and useful for researchers and practitioners interested in sustainable supplier selection approaches and methodologies.

It would be advisable to include the authors' own opinions on future research directions in the analyzed area.

Comments on the Quality of English Language

The English language should be improved when it comes to grammar and style.

Author Response

(The authors gave the same response as above.)

Round 2

Reviewer 2 Report

Comments and Suggestions for Authors

The acronyms list appears twice. My early recommendation was to advertise on the first page that a list exists, not to put it on the first page; 

Figure 1 is ill-located. It is not usual to have exhibits in the first section, as it serves to help the reader decide if she or he will read the rest. Reasoning must go further. I suggest splitting section two (Material and methods) in 2.1 (research Methodology, in which you carefully explain what you did) and 2.2 (Related studies);

The exponential model of Figure 5 does not fit. It would help if you informed the R2 to evaluate the quality of the fitting. In the graph, the y-axis must start from zero to communicate the size, not only the variation of the measure. 

The last chapter is still poorly written, Please insert a critical analysis of what you did and what´s next.

when you state ... The reason is that the index is SCI/SCI-E, included in the Q1 and Q2 ranks, the impact factor is high, and other authors highly cite it. This information is in Table 7. ... it is a tautology. The impact factor is high BECAUSE other authors cite it.

English must be improved, please try proofreading with a native speaker 

Comments on the Quality of English Language

Major review and proofreading

Author Response

Dear Reviewer,

We have attached the response letter 2. Please consider it.

Best regards,

Round 3

Reviewer 2 Report

Comments and Suggestions for Authors

ok